# Hexarelin Modulation of MAPK and PI3K/Akt Pathways in Neuro-2A Cells Inhibits Hydrogen Peroxide—Induced Apoptotic Toxicity

**DOI:** 10.3390/ph14050444

**Published:** 2021-05-08

**Authors:** Ramona Meanti, Laura Rizzi, Elena Bresciani, Laura Molteni, Vittorio Locatelli, Silvia Coco, Robert John Omeljaniuk, Antonio Torsello

**Affiliations:** 1School of Medicine and Surgery, University of Milano-Bicocca, 20900 Monza, Italy; r.meanti@campus.unimib.it (R.M.); laura.rizzi@unimib.it (L.R.); laura.molteni@unimib.it (L.M.); Vittorio.locatelli@unimib.it (V.L.); silvia.coco@unimib.it (S.C.); antonio.torsello@unimib.it (A.T.); 2Department of Biology, Lakehead University, Thunder Bay, ON P7B 5E1, Canada; romeljan@lakeheadu.ca

**Keywords:** hexarelin, GHS, neuroprotection, apoptosis, hydrogen peroxide, oxidative stress

## Abstract

Hexarelin, a synthetic hexapeptide, exerts cyto-protective effects at the mitochondrial level in cardiac and skeletal muscles, both in vitro and in vivo, may also have important neuroprotective bioactivities. This study examined the inhibitory effects of hexarelin on hydrogen peroxide (H_2_O_2_)-induced apoptosis in Neuro-2A cells. Neuro-2A cells were treated for 24 h with various concentrations of H_2_O_2_ or with the combination of H_2_O_2_ and hexarelin following which cell viability and nitrite (NO_2_^−^) release were measured. Cell morphology was also documented throughout and changes arising were quantified using Image J skeleton and fractal analysis procedures. Apoptotic responses were evaluated by Real-Time PCR (caspase-3, caspase-7, Bax, and Bcl-2 mRNA levels) and Western Blot (cleaved caspase-3, cleaved caspase-7, MAPK, and Akt). Our results indicate that hexarelin effectively antagonized H_2_O_2_-induced damage to Neuro-2A cells thereby (i) improving cell viability, (ii) reducing NO_2_^−^ release and (iii) restoring normal morphologies. Hexarelin treatment also reduced mRNA levels of caspase-3 and its activation, and modulated mRNA levels of the BCL-2 family. Moreover, hexarelin inhibited MAPKs phosphorylation and increased p-Akt protein expression. In conclusion, our results demonstrate neuroprotective and anti-apoptotic effects of hexarelin, suggesting that new analogues could be developed for their neuroprotective effects.

## 1. Introduction

Growth hormone secretagogues (GHS) are a class of synthetic oligopeptides and non-peptidyl molecules endowed with endocrine and extra-endocrine properties. GHS preferentially recognize and bind the ghrelin receptor, known as growth hormone secretagogue receptor type-1a (GHS-R1a) [1]. The GHS-R1a is predominantly found in the hypothalamus and pituitary gland where it mediates the release of growth hormone (GH). The GHS-R1a is also implicated in regulation of gastrointestinal motility, as well as energy and glucose homeostasis [2].

In addition to their endocrine effects, GHS also target peripheral tissues; to illustrate, GHS improve muscle function in several pathological conditions by inhibiting the apoptosis pathway, reducing nitric oxide (NO) production, and counteracting inflammation [3,4].

Hexarelin (a synthetic hexapeptide) binds not only to the GHS-R1a but also the CD36 receptor and manifests varied beneficial effects in diseases associated with muscle wasting [5,6,7], chronic heart failure [8], excitotoxicity, neurological disorders, epilepsy and diabetes [9]. Despite the emerging biological importance of hexarelin, its signalling mechanisms have been only partially elucidated. Some studies have demonstrated that hexarelin modulates activation of different intracellular pathways, such as mitogen-activated protein kinases (MAPKs) and phosphoinositide 3-kinase (PI3K)/protein kinase B (Akt) [10,11], and, thereby, could indirectly influence intracellular calcium (Ca^2+^) concentrations [12]. Furthermore, hexarelin protects cells in vitro from apoptosis by inhibiting NO synthesis and reactive oxygen species (ROS) release, modulating caspases activity as well as the expression of proteins belonging to the BCL-2 family [6,12,13,14,15,16]. 

Oxidative stress is involved in the progression of many neuronal disorders, such as Alzheimer’s disease (AD), Parkinson’s disease (PD), Huntington’s disease (HD), and amyotrophic lateral sclerosis (ALS), as well as in cancer, diabetes, and aging [17,18,19].

Hydrogen peroxide (H_2_O_2_) is the most abundant ROS generated through oxidative stress in mitochondria [20]. H_2_O_2_ is formed by dismutation of superoxide radical anions catalysed by superoxide dismutase (SOD); it is freely soluble in aqueous solutions and easily penetrates biological membranes [21]. The diffusibility of H_2_O_2_ in intercellular fluids and the intracellular space can enhance mitochondria damage and potentiate the intrinsic apoptotic pathway [21,22,23]. 

The neuroprotective and anti-apoptotic effects of hexarelin have not yet been assessed. Therefore, the primary aims of this study are to characterize the effects of hexarelin on H_2_O_2_-induced stress and to explore its neuroprotective mechanisms of action in a mouse neuroblastoma cell line (Neuro-2A cells), a well-established in vitro model, frequently used to study the neuroprotective activities of new drugs [24,25,26].

## 2. Results

### 2.1. Effects of Hexarelin on H_2_O_2_-Induced Toxicity in Neuro-2A Cells

Neuro-2A cells were treated with increasing concentrations of H_2_O_2_ (50–200 µM) for 24 h in order to assess its effects on cell replication and to identify the lower concentration that significantly and reproducibly inhibited cell growth. H_2_O_2_ reduced cell replication in a concentration-dependent manner (Figure 1A). As H_2_O_2_ at 100 µM significantly and reproducibly decreased cell replication (*p* < 0.001) compared with control, it was used in subsequent experiments. The effects of H_2_O_2_ were also confirmed by morphological observation of cells using Motic AE2000 Inverted Microscope. Neuro-2A cells exposed to H_2_O_2_ exhibited loss of confluence, disappearance of neurites, as well as grouping and shrinkage that were more marked with increasing H_2_O_2_ concentrations (Figure 1B).

Exposure of Neuro-2A cells to various concentration of hexarelin (10 Nm–10 µM) alone for 24 h did not reduce cell viability; consequently, 1 µM hexarelin was used in subsequent experiments (Figure 2A). The viability of cells treated with the combination of H_2_O_2_ and hexarelin for 24 h was similar to that of control but significantly (*p* < 0.001) greater than that of cells treated with H_2_O_2_ alone (Figure 2B).

### 2.2. Effects of Hexarelin on NO_2_^−^ Production on H_2_O_2_-Induced Neuro-2A Cells

Nitric oxide (NO) is a highly reactive cytotoxic free radical, which can be induced by oxidative stress. Extracellular nitrite (NO_2_^−^) concentrations, proportional to NO formation induced by H_2_O_2_, were measured by the Griess assay. 

Treatment of cells with 1 µM hexarelin alone had no significant effect on NO_2_^−^ release compared with control; by comparison, H_2_O_2_-treatment (100 µM) alone significantly increased levels of NO_2_^−^ (*p* < 0.05) compared with control (Figure 3). In sharp contrast, extracellular NO_2_^−^ levels in cells co-incubated with 1 µM hexarelin and 100 µM H_2_O_2_ were significantly (*p* < 0.05) lower than in cells treated with H_2_O_2_ alone.

These results suggest that the anti-apoptotic properties of hexarelin may be mediated by a presumptive antioxidant mechanism.

### 2.3. Effects of Hexarelin on Morphological Changes Induced by H_2_O_2_ Treatment

Neuro-2A cells were stained as described in Materials and Methods and observed with a confocal laser-scanning microscope (LSM 710, ZEISS, Jena, Germany) in order to characterize morphological changes induced by treatments. A representative photomicrograph for each treatment is presented in Figure 4A. First, we quantified the number of Neuro-2A cells in a fixed area (Figure 4B) by the use of a specific macro for ImageJ software [27]. As shown in Figure 4B, H_2_O_2_ induced a significant reduction in the number of cells per field (*p* < 0.001) compared with control; by comparison, cell numbers were significantly greater when treated with the combination of H_2_O_2_ and hexarelin 1 µM (*p* < 0.05).

The number of cells in each field was used for normalizing data of skeleton analysis, which in turn was used to quantify endpoints and process length [28,29], since the loss of ramifications is a typical characteristic of morphological cytoskeletal changes in apoptosis. Briefly, Analyze Skeleton Plugin was applied to skeleton images obtained after a series of ImageJ plugin protocols of original photomicrographs, as described in Material and Methods, and shown in Figure 12. Process length (Figure 12B, orange) indicate the measure of processes elongation, while endpoints (Figure 12B, blue) are the termination of cellular ramifications. 

Figure 4C,D reveal that H_2_O_2_ treatment alone caused both a reduction of cellular process endpoints and a reduction in summed process length per cell compared with control (*p* < 0.001 and *p* < 0.01, respectively); by comparison, these effects of H_2_O_2_ alone were significantly antagonized by hexarelin co-incubation (*p* < 0.001 and *p* < 0.05, respectively).

We used FracLac for ImageJ to investigate and quantify morphological changes of Neuro-2A cells as a consequence of treatments. Examples of cropped photomicrographs, binary and outline transformations are shown in Figure 5.

Fractal dimension (D) is an index of cell complexity pattern which is used to identify cellular forms ranging from simple rounded to complex branched (Figure 13B) [29,30]. As shown in Figure 6A, H_2_O_2_ significantly decreased D (*p* < 0.001) compared with the control, indicating a reduced branch complexity, according to skeleton analysis. Conversely, Neuro-2A cells treated with the combination of H_2_O_2_ with 1 µM hexarelin exhibited a significantly greater D value (*p* < 0.001) compared with cells treated with H_2_O_2_ alone and which was similar to the D value of controls. 

The lacunarity values of cells treated with 100 µM H_2_O_2_ alone, which ranged from 0.12 to 0.38, were significantly smaller than those of the control group. Lacunarity is a property of the soma based on the heterogeneity or translational and rotational invariance in a shape (Figure 13C); lower lacunarity values indicate a loss of shape heterogeneity [29,30]. Hexarelin antagonized the effects of H_2_O_2_, since cells treated with their combination showed values ranging 0.33 to 0.67, significantly greater (*p* < 0.001) compared to those treated with H_2_O_2_ alone (Figure 6B).

Finally, we analyzed Neuro-2A cell size by: (i) the maximum span across the convex hull (MSACH), which is the maximum distance between two points across the convex hull (Figure 13E); (ii) perimeter, calculated as the number of pixels on the outline cell shape (Figure 13D); and (iii) area, quantified as the total number of pixels present in the filled shape of cell image (Figure 13A “Binary”).

H_2_O_2_ treatment alone significantly reduced MSACH, perimeter and area (Figure 6C–E; *p* < 0.001 for all); this reduction was attenuated by co-incubation with hexarelin. 

The morphological results obtained with skeleton and fractal analysis in a 3D scatter plot are summarized in Figure 7. We have chosen to represent endpoints/cell, fractal dimension and lacunarity because they are independent variables and allow better appreciation of apoptosis-induced cytoskeletal changes.

### 2.4. Effects of Hexarelin on Caspases-3 and -7 and on BCL-2 Family mRNA Levels 

Caspase-3 and caspase-7 activation after damage induced by H_2_O_2_ occur at an early stage of apoptotic cell death; therefore, we hypothesized that hexarelin could inhibit caspase-3 and caspase-7 mRNA levels. 

First, we demonstrated that mRNA levels of both caspases were increased by H_2_O_2_ in a dose-dependent manner and that 100 µM H_2_O_2_ induced a significant increase in caspase-3 (*p* < 0.01) and caspase-7 (*p* < 0.001) mRNA levels (Figure 8A,B). Hexarelin alone did not affect mRNA levels of both caspases (Figure 8C,D). Notably, 1 µM hexarelin antagonized the increase in caspase-3 mRNA levels induced by 100 µM H_2_O_2_ (Figure 8D), whereas it did not oppose the effects of 100 µM H_2_O_2_ on caspase-7 mRNA levels (*p* < 0.001, Figure 8C). 

Moreover, we also quantified the cellular content of activated caspase-3 and -7 proteins; both species are effector caspases, which are activated through proteolytic processing by upstream caspases to produce the mature subunit. Western blot analysis showed that H_2_O_2_ treatment significantly increased cleaved caspase-3 and cleaved caspase-7 (*p* < 0.001) (Figure 9A,B), whereas co-incubation with hexarelin blunted these effects only for caspase-3 activation (*p* < 0.05) (Figure 9A).

Mitochondria also play a crucial role in the process of cell apoptosis with the activation of the BCL-2 protein family. H_2_O_2_, in a concentration-dependent manner, caused a significant increase in the pro-apoptotic Bax mRNA levels (Figure 10A) and tended to increase levels of the anti-apoptotic Bcl-2 mRNA (Figure 10B). Co-incubation for 24 h with hexarelin and H_2_O_2_ reveal the anti-apoptotic effects of hexarelin (Figure 10C,D). The mRNA levels of Bax (stimulates by H_2_O_2_) were significantly (*p* < 0.05) reduced by hexarelin co-incubation; by contrast, hexarelin co-incubation significantly (*p* < 0.01) increased Bcl-2 mRNA levels.

### 2.5. Effects of Hexarelin on ERK 1/2, p38 and Akt Protein Levels in H_2_O_2_-Treated Neuro-2A Cells

We hypothesized that hexarelin could modify MAPK signalling as well. Among members of the MAPK family, ERK and p38 are known to be associated with cell death or survival, respectively [10].

Compared to the control group, exposure to 100 µM H_2_O_2_ alone significantly increased the p-ERK/t-ERK ratio (*p* < 0.05), whereas 1 µM hexarelin alone did not affect ERK protein levels. Notably, co-incubation with hexarelin and H_2_O_2_ induced a trend toward a reduction of p-ERK protein levels compared to the H_2_O_2_ alone group (Figure 11A).

Incubation with 100 µM H_2_O_2_ alone significantly increased the p-p38/t-p38 ratio compared to controls, and this effect was significantly antagonized by co-incubation with hexarelin (*p* < 0.05) (Figure 11B).

Furthermore, levels of p-Akt, associated with cell survival after oxidative stress [31], were significantly reduced by H_2_O_2_ treatment (*p* < 0.05, Figure 11C). Notably, p-Akt levels in cells coincubated with hexarelin and H_2_O_2_ were significantly (*p* < 0.01) greater than those in cells treated with H_2_O_2_ alone. 

## 3. Discussion

Our in vitro findings clearly demonstrate that hexarelin protects Neuro-2A cells from H_2_O_2_-induced damage. 

Oxidative stress arises when the balance between oxidants and antioxidants is disrupted in favour of the former, resulting in potential damage to the organism [21,32]. Although each neurodegenerative disease (NDD) has its own distinct aetiology and differentially affects brain regions, NDDs share elements of oxidative stress, free radical generation and mitochondrial changes, leading to apoptosis [33]. 

Apoptosis is known to be one of the most sensitive biological markers for evaluating oxidative stress caused by imbalance between ROS generation and efficient activity of antioxidant systems [34,35]. Apoptotic cell death is an active process initiated by genetic programs and culminating in DNA fragmentation, characterized by morphological changes, including cell shrinkage, formation of membrane-packaged inclusions, called apoptotic bodies [23], activation of caspases, nucleases, inactivation of nuclear repair polymerases [36], and finally condensation of nuclei [37].

There are numerous inducers of oxidative stress that are capable of causing cytotoxicity and apoptosis in in vitro models. In our study, we used H_2_O_2_ because it is an established method for potency measurement of neuroprotective agents candidates [38,39].

In this study, H_2_O_2_ induced neuronal cytotoxicity in Neuro-2A cells in a dose-dependent manner, as demonstrated by both MTT assay and morphological observations, while increasing concentrations of hexarelin, which by itself did not affect cell viability. In order to determine the protective effects of GHS against H_2_O_2_-induced cytotoxicity, Neuro-2A cells were treated with 1 µM hexarelin. As expected, cell viability of the hexarelin treated group was similar to control cells; by contrast, H_2_O_2_-induced cell death was significantly attenuated by hexarelin.

The protective effects of hexarelin were further confirmed by Griess assay. Excessive levels of NO, an important mediator of cellular communication, implicated in the pathogenesis of NDDs [40] and in caspase-dependent cellular death [41], could be quantified by the measurement of extracellular NO_2_^−^, a primary stable product of NO breakdown. Previous studies reported that H_2_O_2_-induced insult resulted in increased production of NO in neuronal and glial cells [42,43] through the induction of inducible nitric oxide synthase. In this study, we demonstrated that in Neuro-2A cells H_2_O_2_ increased extracellular NO_2_^−^ release, an effect that was blunted by the coincubation with hexarelin. The ability of hexarelin to reduce NO_2_^−^ release suggests that its protective effects against H_2_O_2_ oxidative stress could be mediated through the modulation of apoptosis and downstream pathways.

Moreover, H_2_O_2_-stimulation of Neuro-2A cells induced morphological changes characteristic of an apoptotic phenotype, including de-ramification and reduction of process length, loss of cellular complexity and shape, as well as reduction of cell size. Both skeleton and fractal analysis suggested that hexarelin preserve the cellular complexity, ramification, dimension, heterogeneity and shape comparable to values observed in the control group.

Hexarelin is a synthetic hexapeptide ligand of the GHS-R1a, which is chemically more stable and functionally more potent when compared with ghrelin [44]; consequently, these characteristics make hexarelin a promising alternative to ghrelin [45]. Ghrelin has been demonstrated to protect several cell types such as adipocytes [46], osteoblasts [47], cardiomyocytes and endothelial cells [48] by inhibiting apoptotic stimuli. As well, ghrelin has been shown to have protective effects in vivo, in rats exposed to status epilepticus [49] or in a rat model of cisplatin-induced cachexia [4], and to promote neurogenesis [50].

Similarly to ghrelin, hexarelin has been shown to stimulate cell proliferation of adult hippocampal progenitor (AHP) cells and to protect against growth factor deprivation-induced apoptosis and necrosis [51], principally through the activation of PI3K/Akt pathway [50,52]. Interestingly, hexarelin also blunts the inflammatory process activated by neurodegenerative diseases, stroke, and tumor invasion [53] largely by modulating the release of pro-inflammatory mediators such as cytokines, reactive oxygen species, free radical species and nitric oxide, which could contribute to both neuronal dysfunction and cell death [54,55].

Hexarelin also exert cardioprotective effects, attenuating cardiomyocyte hypertrophy and apoptosis [56], and attenuates mitochondrial abnormalities reported in cancer cachexia [5,57], inducing biogenesis, mitochondrial mass and dynamics restoration, reducing expression of autophagy-related genes and ROS production. This study demonstrates anti-apoptotic effects of hexarelin via the inhibition of caspases activation. Neuro-2A cells treated for 24 h with increasing concentrations of H_2_O_2_ showed a significant activation of caspase-3 and -7. Treatment of cells with 1 µM hexarelin attenuated primarily the activation of caspase-3, both in terms of mRNA levels and protein activation. Our hypothesis was that the modulation of caspase mRNA levels induced by hexarelin was dependent on the intracellular pro-apoptotic signalling molecules belonging to the BCL-2 family.

The BCL-2 family consist of two groups of mediators including (i) anti-apoptotic group mainly represented by Bcl-2, and (ii) a pro-apoptotic group, represented principally by Bax. Both groups play important roles in mitochondrial related apoptosis pathways [58]. 

Therefore, we quantified, by RT-PCR, the effects of hexarelin on modulation of the expression in Bcl-2 and Bax mRNA levels. As expected, H_2_O_2_ treatment of Neuro-2A cells induced a concentration-dependent activation of pro-apoptotic Bax, and the inhibition of anti-apoptotic Bcl-2. Hexarelin treatment did not affect mRNA levels of apoptotic signalling molecules compared to the control group, demonstrating that hexarelin does not stimulate the apoptosis pathway. At the same time, the decrease in Bax mRNA levels and the increase in Bcl-2 mRNA levels, in the group incubated with the combination of hexarelin and H_2_O_2_ confirmed the anti-apoptotic effect of hexarelin.

In order to investigate the molecular pathways involved in hexarelin neuroprotection, we quantified the expression of MAPKs (ERK and p38) and PI3K/Akt. MAPKs activation contribute to neuronal dysfunction and are involved in NDDs [59,60]. Furthermore, ERK has been shown to participate in the regulation of cell growth and differentiation, and responses to cellular stress [61].

In this study, treatment of Neuro-2A cells with H_2_O_2_ led to cell death by up-regulating p-ERK and p-p38 protein expression. The up-regulation of MAPKs induced by H_2_O_2_ stimulation were blunted by hexarelin treatment. In addition, hexarelin alone did not affect the MAPKs proteins compared to the control.

PI3K/Akt is a key anti-apoptotic effector in the growth factor signalling pathway [62]. In particular, the phosphorylation of Thr-308 and Ser-473 of Akt serves a key role in mediating the anti-apoptotic actions of growth factors on cells and plays an important role in neuronal protection [63,64]. In this study, H_2_O_2_-induced oxidative stress significantly increased the de-phosphorylation of Akt, which stimulated the activation of the apoptotic pathway. Hexarelin treatment did not alter the p-Akt/t-Akt ratio compared to controls, but in cells treated for 24 h with both hexarelin and H_2_O_2_, Akt protein levels were significantly increased compared with cells treated with H_2_O_2_ alone.

In conclusion, our findings demonstrate that H_2_O_2_ caused early and late apoptotic pathways in Neuro-2A cells. Treatment of cells with hexarelin antagonized H_2_O_2_ cellular cytotoxicity, inhibited apoptosis and potentiated MAPKs and PI3K/Akt survival pathways.

This study demonstrates that hexarelin is capable of protecting Neuro-2A cells from H_2_O_2_-caused cytotoxicity effects; however, further investigations are required to clarify hexarelin molecular mechanisms of action, and whether its effects are mediated by the ghrelin receptor (GHS-R1a).

## 4. Materials and Methods

### 4.1. Chemicals

Hexarelin, Dulbecco’s Modified Eagle’s Medium (DMEM)-high glucose, hydrogen peroxide (H_2_O_2_), 3 (4,5 dimethylthiazol-2yl)-2,5-diphenyl tetrazolium bromide (MTT), Griess reagent, poly-D-lysin hydrobromide, 4’,6-diamidino-2-phenylindole dihydrochloride (DAPI), fluoromount aqueous mounting medium and bovine serum albumin (BSA) were purchased from Sigma-Aldrich (St. Louis, MO, USA). Penicillin, streptomycin, L-glutamine, trypsin-EDTA, phosphate-buffer saline (PBS) and fetal bovine serum (FBS) were obtained from Euroclone (Pero, Milan, Italy). Alexa Fluor 488 Phalloidin was purchased by ThermoFisher Scientific (Waltham, MA, USA). Prior to assay, hexarelin was dissolved in ultrapure water; both hexarelin and H_2_O_2_ were diluted in culture medium to final working concentrations.

### 4.2. Cell Culture

Immortalized Neuro-2A murine neuroblastoma cells were obtained from Interlab Cell Line Collection (ICLC, Genoa, Italy) and cultured in DMEM-high glucose (Sigma-Aldrich, St. Louis, MO, USA) supplemented with 10% heat-inactivated FBS, 100 IU/mL penicillin, 100 μg/mL streptomycin and 2 mM L-glutamine (all Euroclone, Pero, Italy) and Mycozap Prophylactic (Lonza, Walkersville, MD, USA) under standard cell culture conditions (37 °C, 5% CO_2_) [24,25]. Confluent cultures were washed with PBS, detached with trypsin-EDTA solution (all Euroclone, Pero, Italy), and used for experiments. 

In each experiment, Neuro-2A cells were incubated with H_2_O_2_ alone or the combination of 100 µM H_2_O_2_ and 1 μM hexarelin for 24 h. 

### 4.3. Cell Viability

Neuro-2A cells were seeded in 96-well culture plates at the density of 4 × 10^4^ cells/well and cultured for 24 h at 37 °C. The day after seeding, the cells were treated with increasing concentrations (50–200 µM) of H_2_O_2_ or hexarelin (10 nM–10 µM) or with 100 µM H_2_O_2_ and hexarelin (1 µM). After 24 h of treatment, a 10 µL aliquot of 5 mg/mL MTT (M5655, Sigma-Aldrich, St. Louis, MO, USA) was added to each well and incubated at 37 °C for 3 h. Then, the culture medium was removed and a 200 µL aliquot of acidified isopropanol was added in order to dissolve the formazan crystals. Absorbance was read at 570 nm using a multilabel spectrophotometer VICTOR^3^ (Perkin Elmer, Waltham, MA, USA). Cell viability of control cells was set to 100% and the relative absorbances of experimental groups were converted to relative percentages (relative absorbance of experimental group/relative absorbance of control) × 100 = % of viable cells.

### 4.4. Griess Assay

NO production was evaluated measuring the nitrite (NO_2_^−^) content of culture media with the Griess reaction. Briefly, Neuro-2A cells were plated in 96-well culture plates and treated with 100 µM H_2_O_2_ and 1 µM hexarelin for 24 h. At the end of the treatment, 100 µL aliquots of medium were transferred to a new 96-well plate and were mixed with 100 µL of Griess reagent 1 × (G4410, Sigma-Aldrich, St. Louis, MO, USA). Absorbance was measured at 540 nm with the VICTOR^3^ spectrophotometer (Perkin Elmer, Waltham, MA, USA), after 15 minutes in the dark at room temperature. A standard curve with varied concentrations of sodium nitrite (S2252, Sigma-Aldrich, St. Louis, MO, USA) was conducted in parallel and used for quantification. NO_2_^−^ of control cells was set to 100% and the relative absorbance of experimental groups were converted to relative percentages (relative absorbance of experimental group/relative absorbance of control) × 100 = % of NO_2_^−^. The final nitrite concentration is proportional to the NO metabolite present in the sample.

### 4.5. Observation of Morphological Changes

Neuro-2A cells were seeded in 6-well culture plates at a density of 80 × 10^4^ cells/well and incubated at 37 °C for 24 h. Cells were incubated with various concentrations (50 to 200 µM) of H_2_O_2_ and photographed 24 h later using a Motic AE2000 Inverted Microscope (Motic, Hong Kong, China).

### 4.6. Actin Staining Assay

Neuro-2A cells (2 × 10^5^ cells/well) were seeded on coverslips coated with poly-D-lysine (P0899, Sigma-Aldrich, St. Louis, MO, USA) for 24 h. Cells were incubated at 37 °C with 100 µM H_2_O_2_ for 24 h with or without hexarelin 1 µM, then washed with PBS and fixed with 4% paraformaldehyde (Titolchimica, Rovigo, Italy) for 10 min at room temperature. Cells were subsequently washed with PBS, incubated with cold acetone for 5 min at −20 °C and blocked in PBS with 1% BSA for 30 min at room temperature. In order to stain actin, Neuro-2A cells were incubated with 2 U/mL of Alexa Fluor 488 Phalloidin diluted in PBS with 1% BSA at room temperature for 20 min and then washed with PBS. Counterstaining of nuclei was made with 1 µg/mL of DAPI for 10 min at room temperature. After washing cells with PBS, fluoromount aqueous mounting medium was added, and the cells were observed under a confocal laser scanning microscope (LSM 710, ZEISS, Jena, Germany); images were captured at 40× and 63× magnification by ZEN software (ZEISS, Jena, Germany).

### 4.7. Morphological Analysis

Photomicrographs obtained at 40x magnification were used to evaluate the number of cells in the same area using a specifically designed macro with ImageJ software (National Institutes of Health, Bethesda, MD, USA) [27]. The same photomicrographs were used for skeleton analysis [28]. Skeleton analysis was applied to quantify the number of process endpoints and length normalized by the number of cells in the same area. Briefly, the photomicrographs were filtered to soften the background, enhance the contrast and remove noise, by using the application Fiji free software (https://imagej.net/fiji, accessed on 8 March 2021). All the images were then binarized, subsequently skeletonized and analysed with Analyze Skeleton (2D/3D) plugin (http://imagej.net/AnalyzeSkeleton, accessed on 8 March 2021). The overlay of skeleton to original image is reported to demonstrate that skeletons are representative of the original image; all the modification steps are illustrated in Figure 12.

Moreover, we applied fractal analysis using FracLac plugin for ImageJ (https://imagej.nih.gov/ij/plugins/fraclac/fraclac.html, (accessed on 8 March 2021) in order to evaluate the Neuro-2A shape and morphology by different parameters (fractal dimension, lacunarity, maximum span across the hull, perimeter and area) [30]. Photomicrographs obtained with 63 × oil immersion objective were modified similarly to skeleton analysis. Photomicrographs of cells were cropped and transformed to 8-bit grayscale images. Then, cell images were binarized and manually edited to obtain a single cell made of continuous set of pixels. To avoid bias, this modification was done taking into account the original image. Binary images were outlined and analyzed with Fractal Analysis plugin. All the modification steps and representative images of FracLac box counting analysis are shown in Figure 13.

### 4.8. Real-Time PCR Analysis

In order to monitor the apoptosis pathway, Neuro-2A cells were plated in 24-well culture plates at a density of 2 × 10^5^ cells/well, and treated for 24 h according to specific protocols. Following treatment, Neuro-2A cells were washed with PBS and total RNA was extracted using EuroGOLD Trifast reagent (Euroclone, Pero, MI, Italy), according to the manufacturer’s instructions and quantified using a Nanodrop ND-1000 spectrophotometer (Thermo Fisher Scientific, Waltham, MA, USA). Reverse transcription was performed using iScript™ cDNA Synthesis Kit (Bio-Rad, Hercules, CA, USA) to equal amounts (140 ng) of RNA. Amplification of cDNA (21 ng) was performed in a total volume of 20 µL of iTaq Universal Probes Supermix (Bio-Rad), using Real-Time QuantStudio7 Flex (Thermo Fisher Scientific, Waltham, MA, USA). After 2 min at 50 °C and 10 min at 94.5 °C, 40 PCR cycles were performed using the following conditions: 15 s at 95 °C and 1 min at 60 °C. Relative mRNA concentrations of the target genes were normalized to the corresponding β-actin internal control and calculated using the 2^−ΔΔCt^ method.

### 4.9. Western Blot Analysis

Neuro-2A cells were plated in 6-well culture plates at a density of 8 × 10^5^ cells/well, incubated at 37 °C for 24 h and then treated as previously described. Following treatment, cells were rinsed with ice-cold PBS and lysed in RIPA buffer (Cell Signaling Technology, Danvers, MA, USA), supplemented with a protease-inhibitor cocktail (Sigma-Aldrich, St. Louis, MO, USA), according to the manufacture’s protocol.

Total protein concentrations were determined using the Pierce BCA Protein Assay Kit (Thermo Fisher Scientific, Waltham, MA, USA). Equal amounts of protein (20 µg) were heated at 95 °C for 10 min, loaded on precast 4–12% gradient gels (Invitrogen, Carlsbad, CA, USA), separated by electrophoresis, and transferred to a polyvinylidene difluoride (PVDF) membrane (Thermo Fisher Scientific, Waltham, MA, USA). Non-specific binding was blocked with 5% dried fat-free milk dissolved in phosphate-buffered saline (PBS) supplemented with 0.1% Tween-20 (PBS-T) for 1 h at room temperature. After 3 washes in PBS-T, membranes were incubated with the primary antibody overnight at 4°C (Anti-cleaved caspase-3 (Asp175) (5A1E) rabbit antibody, #9664, Cell Signaling Technology, 1:1000; anti-cleaved caspase-7 (Asp198) rabbit antibody, #9491, Cell Signaling Technology, 1:1000; anti-Phospho-p44/42 MAPK (Erk1/2) (Thr202/Tyr204) rabbit antibody, #9101, Cell Signaling Technology, 1:1000; anti-p44/42 MAPK (Erk1/2) rabbit antibody, #4695, Cell Signaling Technology, 1:1000; anti-Phospho-p38 MAPK (Thr180/Tyr182) rabbit antibody, #4511, Cell Signaling Technology, 1:1000; anti-p38 MAPK rabbit antibody, #9212, Cell Signaling Technology, 1:1000; anti-Phospho-Akt rabbit antibody, #4060, Cell Signaling Technology, 1:2000; anti-Akt rabbit antibody, #4685, Cell Signaling Technology, 1:1000; anti-actin rabbit antibody, #A2066, Sigma Aldrich, 1:2500), and then with a peroxidase-coupled goat anti-rabbit IgG (#31460, Thermo Scientific, 1:5000) for 1 h at room temperature.

Signals were developed with the extra sensitive chemiluminescent substrate LiteAblot TURBO (Euroclone, Pero, Milan, Italy) and detected with Amersham ImageQuant 800 (GE Healthcare, Chicago, IL, USA). Image J software was used to quantify protein bands.

### 4.10. Statistical Analysis

Statistical analysis was performed using the program GraphPad Prism (GraphPad Software, La Jolla, CA, USA). Values are expressed as mean ± standard error of the mean (SEM). Experiments were independently replicated at least three times. One-way ANOVA followed by Tukey’s *t*-test was used for multiple comparisons. A *p*-value of less than 0.05 was considered significant.

## Figures and Tables

**Figure 1 pharmaceuticals-14-00444-f001:**
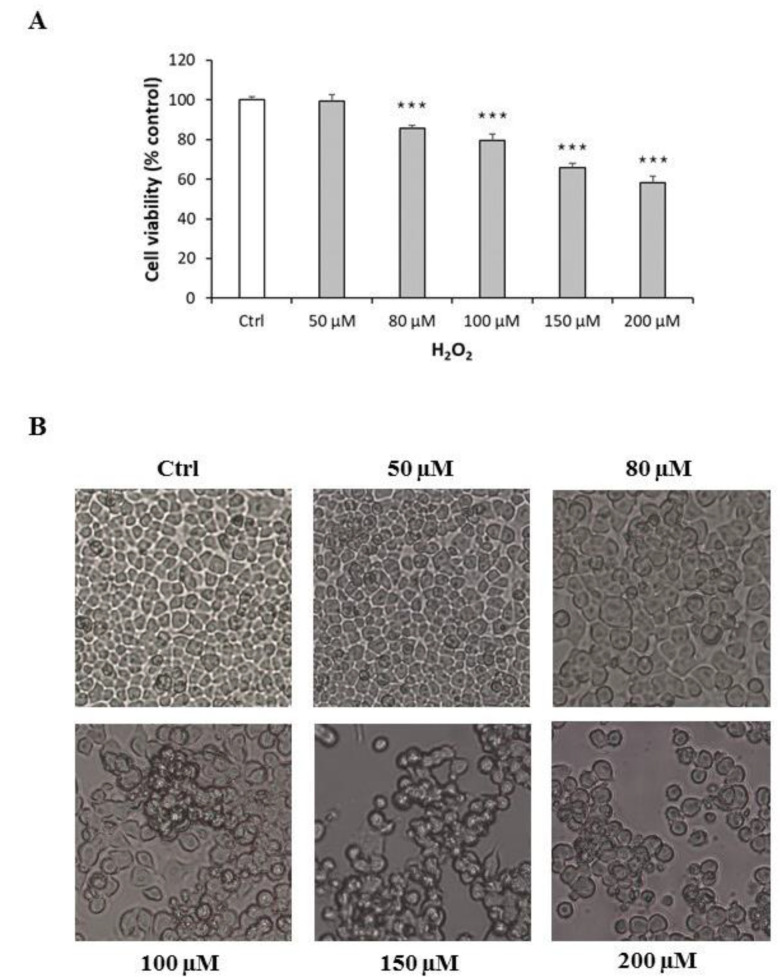
Cytotoxic effect and morphological alterations of Neuro-2A cells in response to increasing concentration of H_2_O_2_. (**A**) Neuro-2A cells were treated for 24 h with different concentrations of H_2_O_2_ (0, 50, 80, 100, 150, 200 μM) and assessed for MTT assay. The assay was performed in at least three independent experiments (*n* = 21). Statistical significance: *** *p* < 0.001 vs. CTRL. (**B**) Morphological changes in Neuro-2A cells were observed by Motic AE2000 Inverted Microscope, at 10× magnification. Representative images of cells treated with increasing concentration of H_2_O_2_.

**Figure 2 pharmaceuticals-14-00444-f002:**
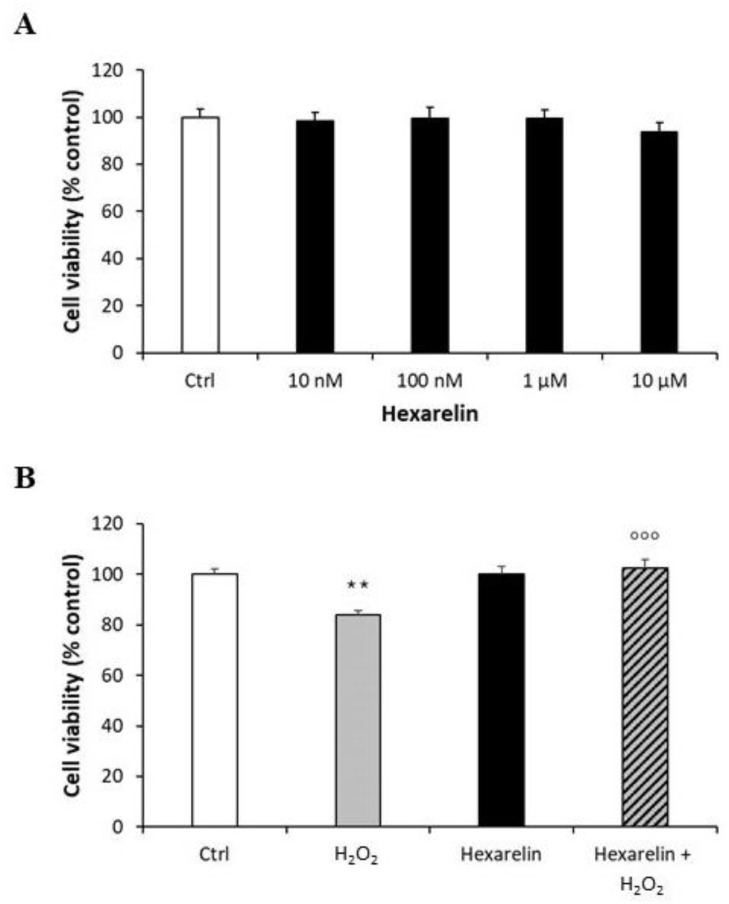
Protective action of hexarelin in Neuro2A cells. (**A**) Neuro-2A cells were treated for 24 h with different concentrations of hexarelin (10 nM, 100 nM, 1 µM, and 10 µM), (**B**) with or without hexarelin (1 µM) and H_2_O_2_ (100 μM) and assessed for MTT assay. All assays were performed in at least three independent experiments (*n* = 21). Statistical significance: ** *p* < 0.01 vs. CTRL; °°° *p* < 0.001 vs. H_2_O_2_.

**Figure 3 pharmaceuticals-14-00444-f003:**
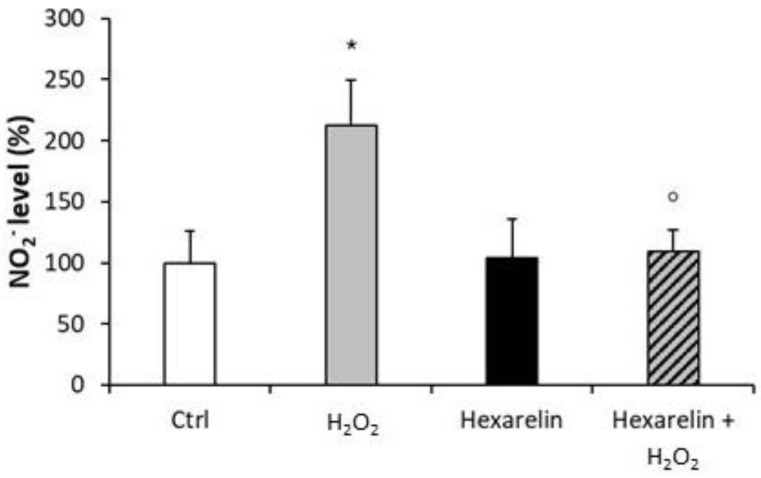
Hexarelin reduced the extracellular NO_2_^−^ release induced by H_2_O_2_. Neuro-2A cells were treated for 24 h with or without hexarelin and 100 µM H_2_O_2_. The culture media were used for Griess reaction to measure NO_2_^−^ extracellular release. Data are expressed as mean ± SEM of 4 replicates (*n* = 24). Statistical significance: * *p* < 0.05 vs. CTRL; ° *p* < 0.05 vs. H_2_O_2_.

**Figure 4 pharmaceuticals-14-00444-f004:**
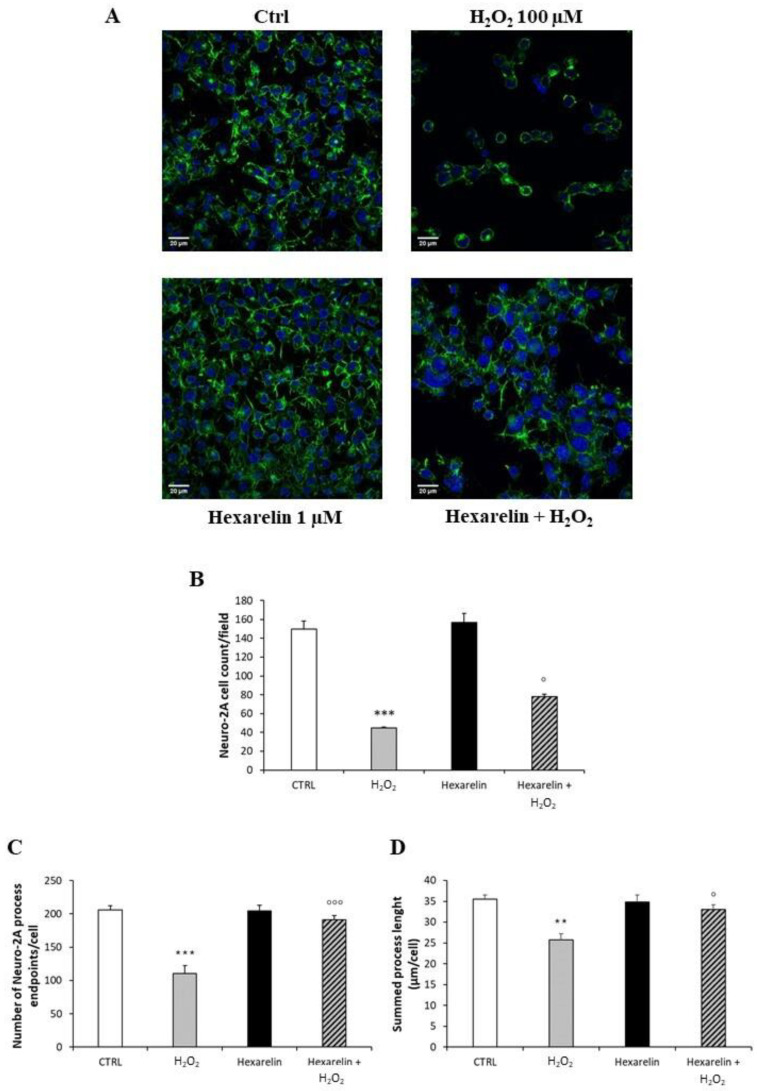
Hexarelin reduces Neuro-2A cells de-ramification induced by H_2_O_2_. (**A**) Neuro-2A cells were seeded on poly-D-lysine pre-treated coverslips and incubated for 24 h with or without hexarelin and 100 µM H_2_O_2_. At the end of the treatment, cells were fixed and stained for phalloidin and DAPI. Images were captured with confocal laser scan microscope. Scale bar: 20 µm. (**B**) Graphical representation of the number of cells in the same areas per each treatment, (**C**) of the process endpoints/cells and (**D**) process length/cells. Data are expressed as mean ± SEM of 3 replicates (total number of cells analyzed = 160). Statistical significance: ** *p* < 0.01, *** *p* < 0.001 vs. CTRL; ° *p* < 0.05, °°° *p* < 0.001 vs. H_2_O_2_.

**Figure 5 pharmaceuticals-14-00444-f005:**
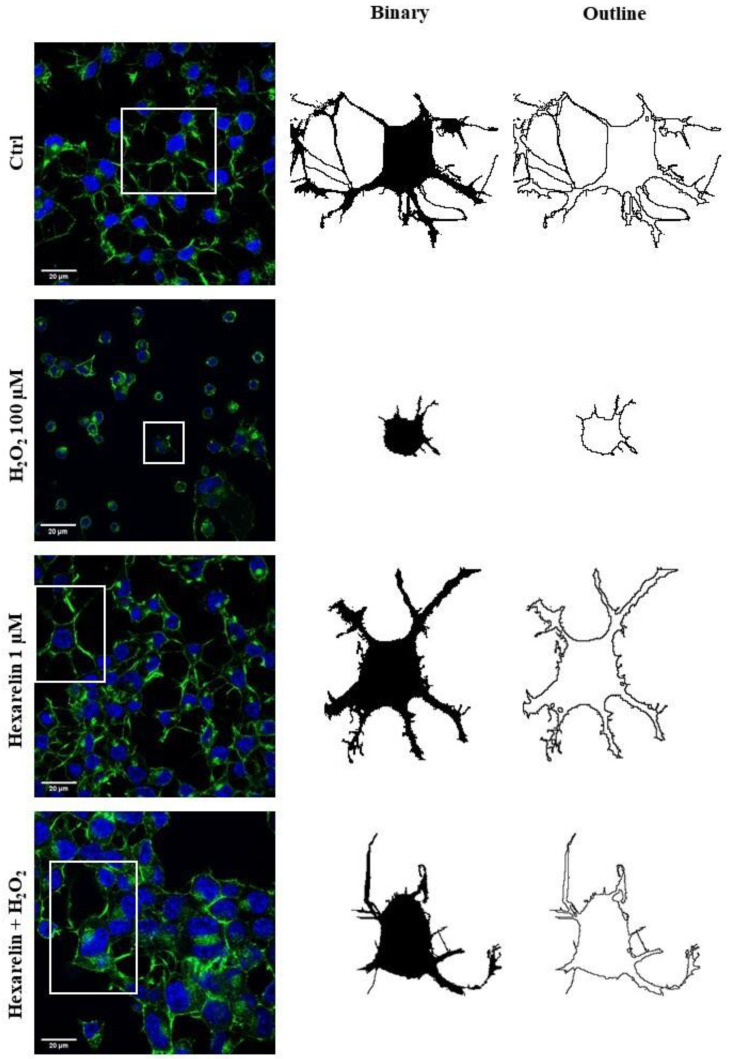
Effects of hexarelin on morphology of Neuro-2A cells stimulated with H_2_O_2_. Representative photomicrographs of Neuro-2A cells incubated for 24 h with or without hexarelin and 100 µM H_2_O_2_, and examples of cell binarized and outlined. Cells were seeded on poly-D-lysine pre-treated coverslips and, at the end of the treatments, were fixed and stained with phalloidin and DAPI. Images were captured with a confocal laser scan microscope. Scale bar: 20 µm.

**Figure 6 pharmaceuticals-14-00444-f006:**
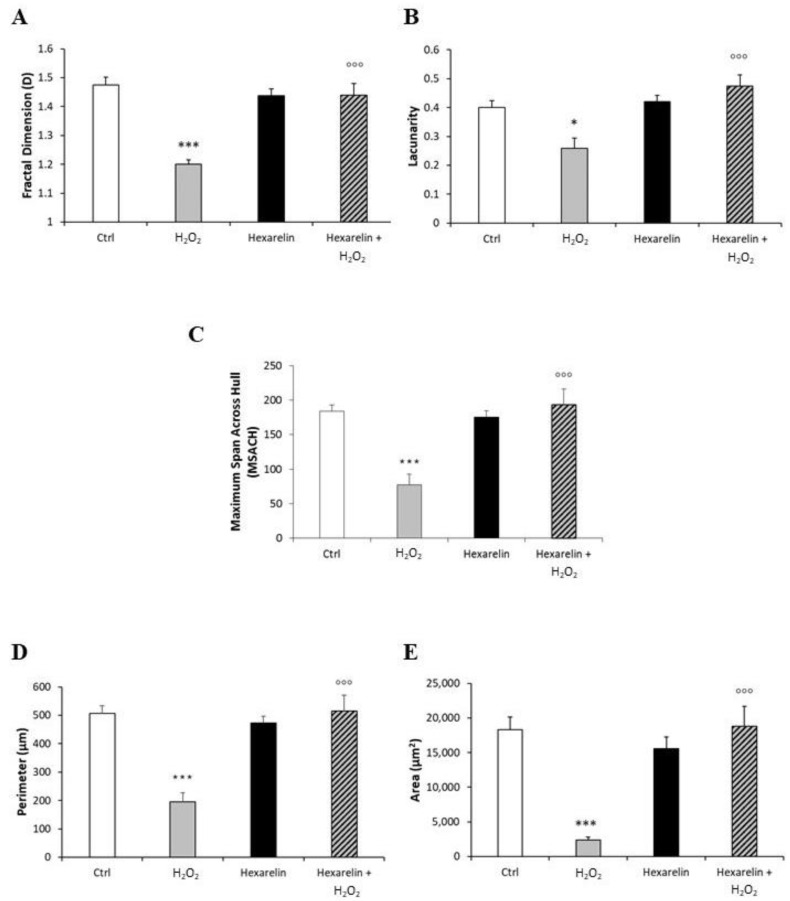
Hexarelin modulates morphological changes of Neuro-2A cells induced by H_2_O_2_ treatment. (**A**) Fractal dimension, (**B**) lacunarity, (**C**) maximum span across hull, (**D**) perimeter, and (**E**) area graphical representation of Neuro-2A cells treated for 24 h with or without hexarelin and 100 µM H_2_O_2_. Total number of cells analyzed for each condition = 10. Data are expressed as mean ± SEM. Statistical significance: * *p* < 0.05, *** *p* < 0.001 vs. CTRL; °°° *p* < 0.001 vs. H_2_O_2_.

**Figure 7 pharmaceuticals-14-00444-f007:**
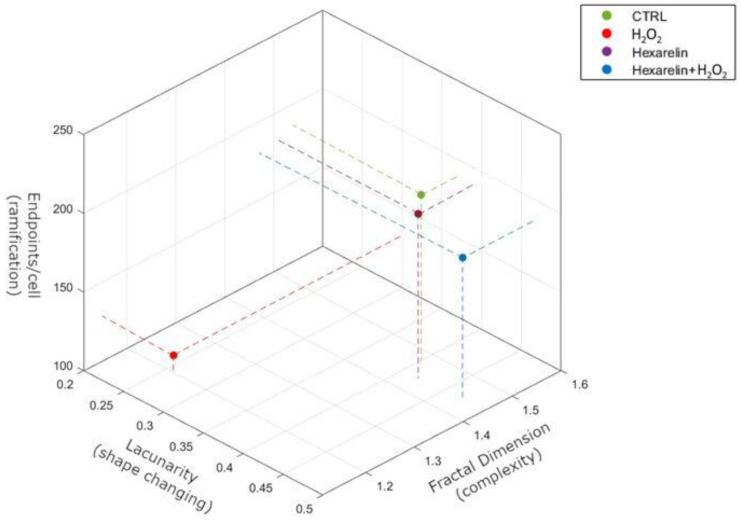
Summary representation of Skeleton and FracLac analysis by 3D scatter plot. The figure summarizes the relationship between ramifications (endpoints/cell), shape changing (lacunarity) and complexity (fractal dimension). Pearson’s correlation demonstrates that fractal dimension significantly correlates with endpoints/cell (r = 0.9806, *p* = 0.0098) but not with lacunarity (r = 0.8134, *p* = 0.0981). Endpoints/cell directly correlates with both fractal dimension (r = 0.9806, *p* = 0.0098) and lacunarity (r = 0.7765, *p* = 0.0038).

**Figure 8 pharmaceuticals-14-00444-f008:**
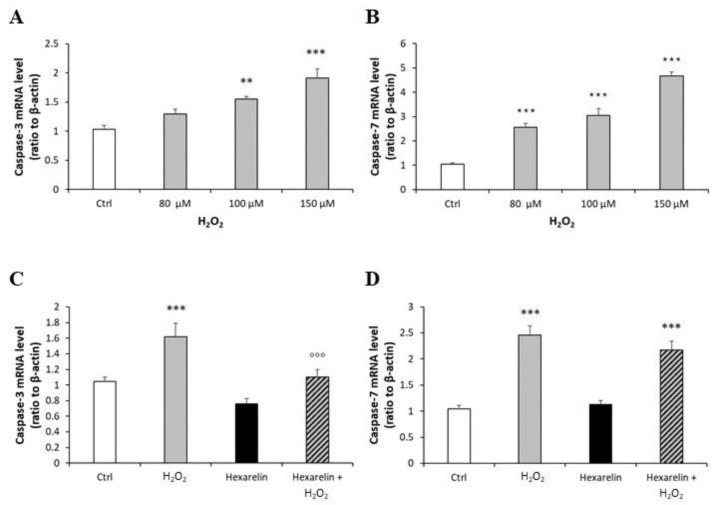
Analysis of caspase-3 and caspase-7 mRNA levels following H_2_O_2_ exposure and their modulation induced by hexarelin. Neuro-2A cells were treated with different concentrations of H_2_O_2_ (80, 100, 150 μM) or co-incubated with 1 µM hexarelin and 100 µM H_2_O_2_ for 24 h. Caspase-3 (**A**,**C**) and caspase-7 (**B**,**D**) mRNA levels were measured by RT-PCR and normalized for the respective β-actin mRNA levels. Data are expressed as mean ± SEM of 3 replicates (*n* = 18). ** *p* < 0.01, *** *p* < 0.001 vs. CTRL; °°° *p* < 0.001 vs. H_2_O_2_.

**Figure 9 pharmaceuticals-14-00444-f009:**
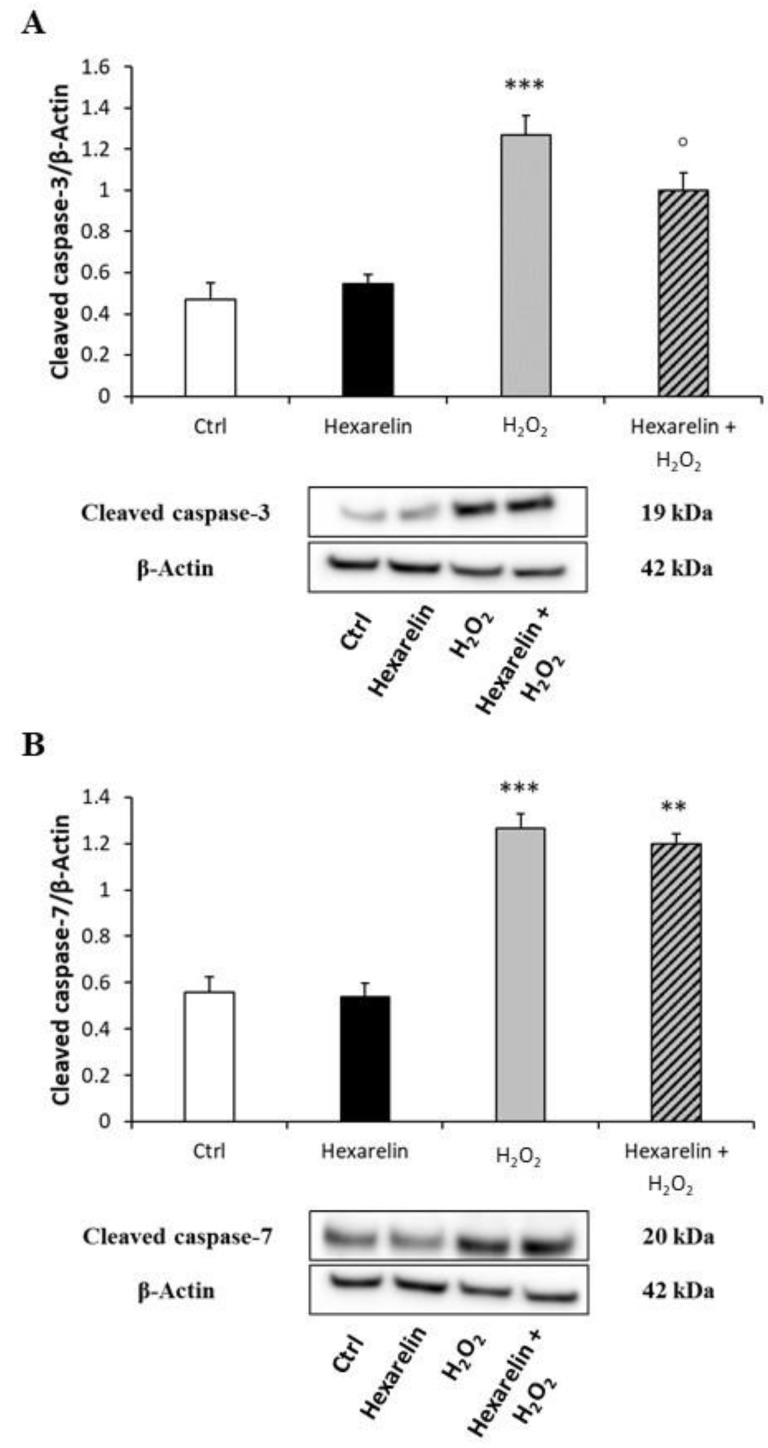
Hexarelin inhibits apoptotic pathway through caspase-3 inactivation. Neuro-2A cells were treated with or without hexarelin and H_2_O_2_ for 24 h and assessed in Western blot for cleaved caspase-3/β-actin (**A**), and cleaved caspase-7/β-actin (**B**). All assays were performed in at least 3 independent experiments (*n* = 3). Statistical significance: ** *p* < 0.01, *** *p* < 0.001 vs. CTRL; ° *p* < 0.05 vs. H_2_O_2_.

**Figure 10 pharmaceuticals-14-00444-f010:**
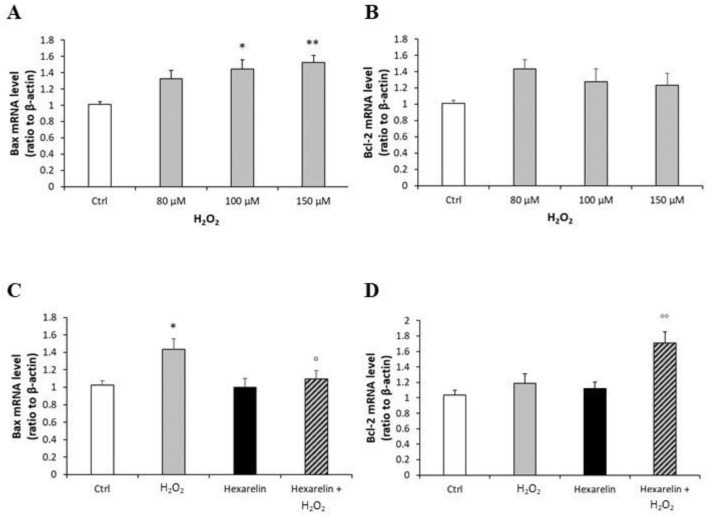
Analysis of mRNA levels of apoptosis markers following H_2_O_2_ exposure and their modulation induced by hexarelin. Neuro-2A cells were treated with different concentrations of H_2_O_2_ (80, 100, 150 μM) or co-incubated with hexarelin 1 µM and H_2_O_2_ 100 µM for 24 h. Bax (**A**,**C**) and Bcl-2 (**B**,**D**) mRNA levels were normalized for the respective β-actin mRNA levels. Data are expressed as mean ± SEM of 3 replicates (*n* = 18). * *p* < 0.05, ** *p* < 0.01 vs. CTRL; ° *p* < 0.05, °° *p* < 0.01 vs. H_2_O_2_.

**Figure 11 pharmaceuticals-14-00444-f011:**
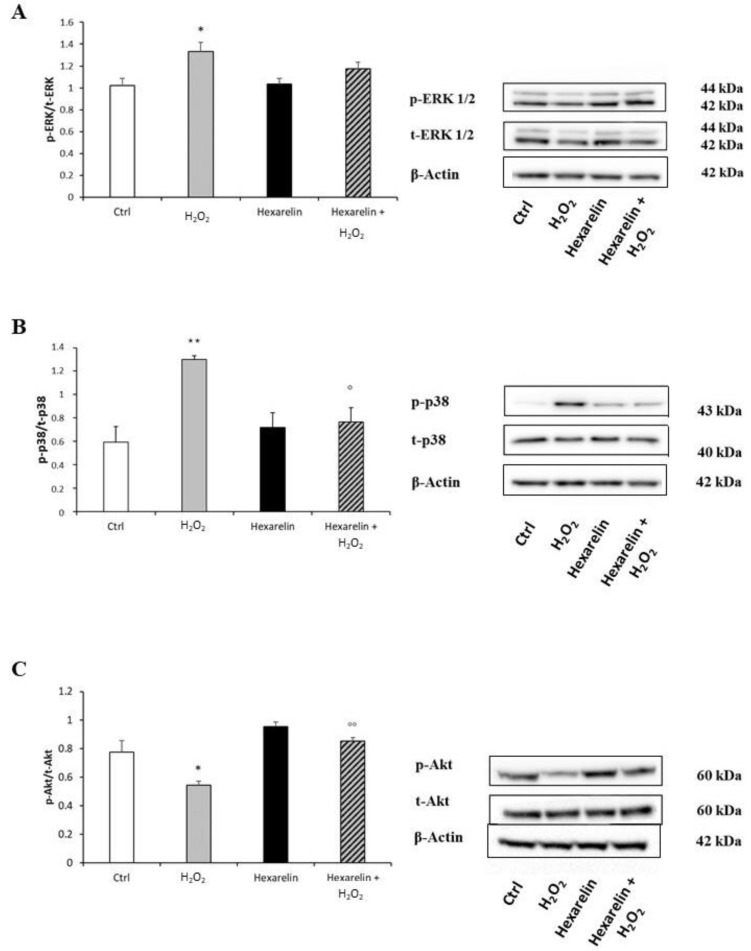
Hexarelin modulates ERK, p38 and Akt activation. Neuro-2A cells were treated with or without hexarelin and H_2_O_2_ for 24 h and Western blot was used to measure p-ERK/t-ERK (**A**), p-p38/t-p38 (**B**), and p-Akt/t-Akt (**C**) levels. β-actin was used to control even protein loading in all lines. All assays were performed at least 3 independent experiments. Statistical significance: * *p* < 0.05, ** *p* < 0.01 vs. CTRL; ° *p* > 0.05, °° *p* < 0.01 vs. H_2_O_2_.

**Figure 12 pharmaceuticals-14-00444-f012:**
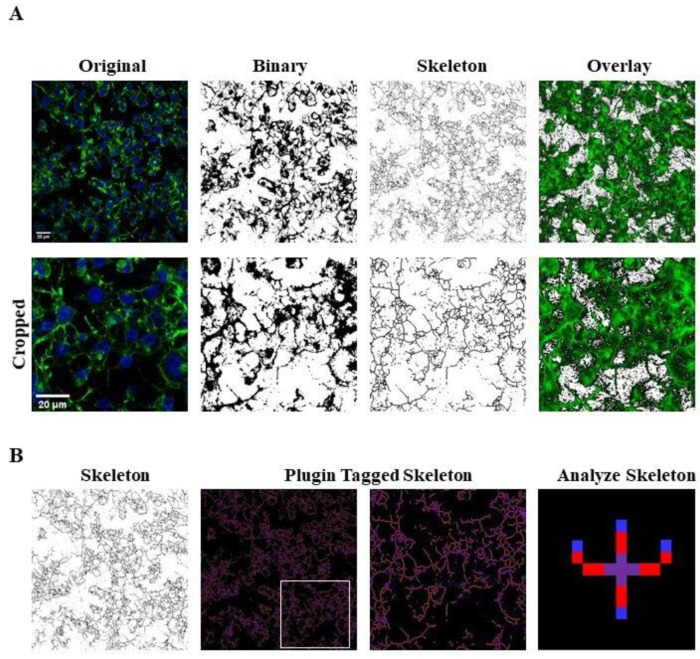
Skeleton Analysis application to quantify Neuro-2A cells morphology. (**A**) Representative photomicrograph (40× magnification) and the series of ImageJ plugin protocols which were applied to each photomicrograph for skeleton analysis. Original photomicrograph was modified enhancing the background, removing noise and using with FFT filter prior to be converted to binary images. Binary image was skeletonized. The overlay of skeletonized and original images is reported; cropped photomicrographs show the image details. Scale bar: 20 µm. (**B**) The workflow used to Analyze Skeleton plugin: skeletonized process in orange, endpoint in blue, and junction in purple.

**Figure 13 pharmaceuticals-14-00444-f013:**
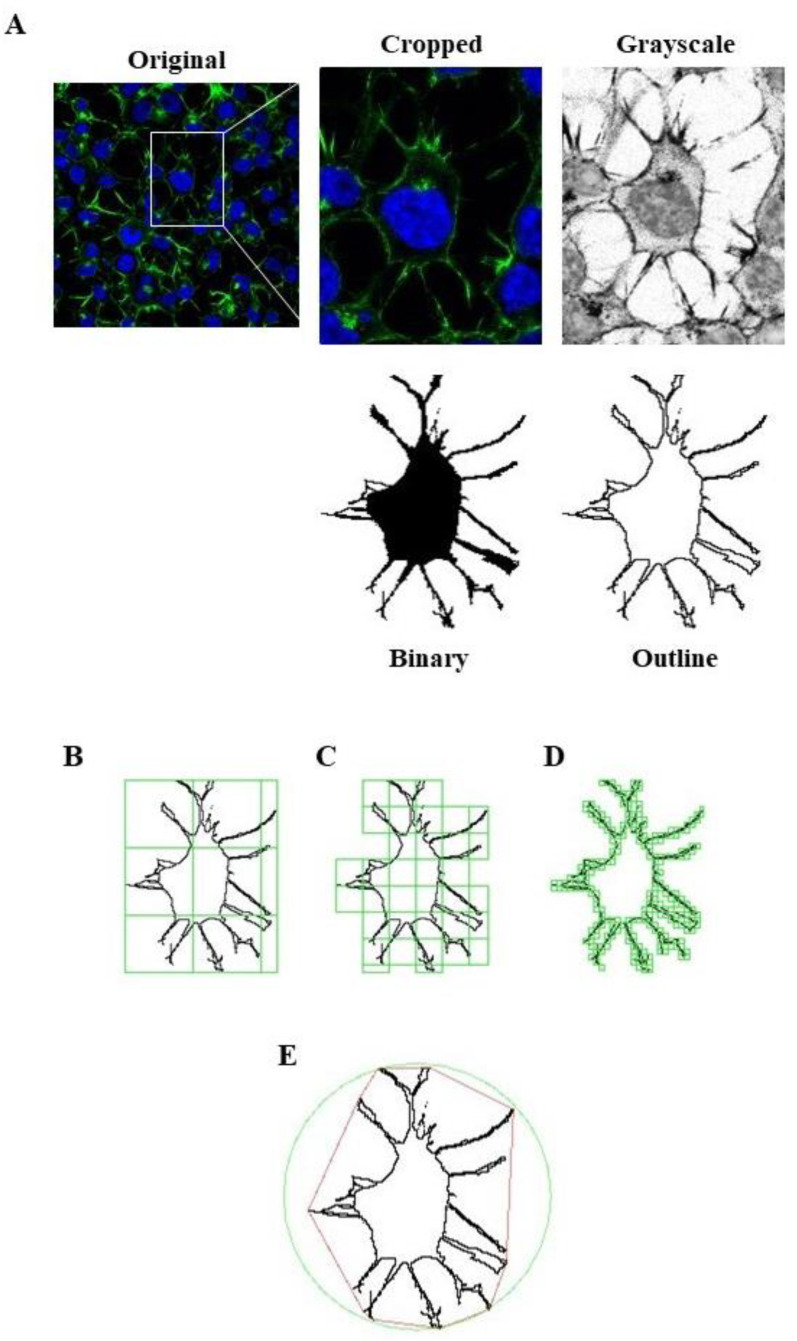
FracLac Analysis application to quantify Neuro-2A cells morphology. (**A**) Representative process applied to obtain an outlined single cell for FracLac plugin. After selecting a cell in the photomicrograph (63× magnification), the image was cropped and modified to remove noise and enhance the background. The image was then processed to obtain an 8-bit grayscale microphotograph, and binarized. Binary image was manually edited to clear the background and to join all branches, and finally outlined. FracLac quantifies cell complexity and shape with a box counting method which permits to quantify fractal dimension (**B**), lacunarity (**C**), perimeter (**D**), and the maximum span across the convex hull (**E**) by drawing a convex hull (pink) and a bounding circle (green).

## Data Availability

Not applicable.

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
