# Peer review of "Hexarelin Modulation of MAPK and PI3K/Akt Pathways in Neuro-2A Cells Inhibits Hydrogen Peroxide—Induced Apoptotic Toxicity"

_pharmaceuticals, 2021, doi:10.3390/ph14050444_

Round 1
Reviewer 1 Report
Meanti et al. demonstrate in this work that hexarelin attenuates apoptosis induced by hydrogen peroxide in Neuro-2A Cells and affect ERC 1/2, p38 and Akt proteins levels. I think it is worth to be published but several issues should be considered:
- Page 2. The authors use 100 μM H2O2 because, at this concentration, cell replication significantly decreased compared with control.
However, the viability of Neuro-2A cells at this concentration is 80% to that of the untreated cells. Why not use 150 uM or 200 uM H2O2? Cell viability at those concentrations drop to 70-60%, and the anti-apoptotic action of hexarelin could be more evident. Have the authors studied the apoptotic-protective effect at those concentrations? Also, the hexarelin concentration used in this work was 1 uM. Why did not the authors use 10 uM? This concentration does not seem to compromise cell viability and could have a higher protective action.
- Page 5: The authors measure several parameters to study morphological changes caused by H2O2 (“The number of cells in each field was used for normalizing data of skeleton analysis, which in turn was used to quantify endpoints and process length”). Although the authors indicate references 28 and 29 for a description of the analysis of the morphological changes, it should be helpful for the reader a brief description of what “endpoints” and “process length” mean. Also, a picture of a real cell showing the parameters analysed in the bar diagrams of figures 4, and 6, should be added (“number o Neuro-2A process endpoints”, “Summed process length”), (Fractal dimensión, lacunarity, máximum spam, perimeter, área).
Maybe Fig 13b and c could be used to indicate how the authors measured each of these parameters.
- Fig. 8. Hexalerin decreases Cas3 mRNA and this may be beneficial for preventing apoptosis but may also have a pro-oncogenic effect. Please discuss.
- In Fig. 9, the bar diagrams and the wester-blot are represented following a different order (Ctrl, H202, Hexarelin and Hexrelin for the bar diagram and Ctrl, Hexalerin, H2O2, Hexalerin+H2O2 for the western blot). Please, use the same order to make the analysis easier.
- Fig. 11 Effects of hexarelin on ERK 1/2, p38 and Akt protein levels in H2O2-treated Neuro-2A cells. The western blots in Fig. 11 need further explanations. Were different western blots used to quantify p-ERK/t-ERK ratios? In this case, a loading control (for instance, alpha-tubulin) should be required to normalize the amount of protein loaded for each sample. The same apply for p-p38/t-p38, and p-Akt/tAkt) ratios.
Do the authors have a model of the hexarelin effect on the mitochondrial apoptotic pathway through the MAPK pathway?
- The number “2” in H2O2 legends should be in subscript types in the bar diagram legends of figures 2B, 3, 4B-D, 6 A-E, 7, 8C-D, 9, 10C-D, and 11A-C.
Author Response
- Many H2O2 concentrations and incubation times have been reported in in vitro experimental models of oxidative stress[1]. Interestingly, the higher the concentration of H2O2 the shorter shortest the incubation time. The low concentration of H2O2 (such as 100 µM) required 24 hours of incubation in order to demonstrate its effectiveness. Accordingly, in our experimental model, which is a simplified model of oxidative stress in neurodegenerative diseases, we have chosen to use H2O2 100 µM for 24 hours. We have not studied the effects of higher concentrations of H2O2 yet, but it is an interesting point for future investigations. The choice of hexarelin at the concentration of 1 µM is based on our long experience in the use of growth hormone secretagogues[2] both in in vivo and in vitro models and also from supporting data from the literature[3].
- We have amended paragraph 3. Effects of hexarelin on morphological changes induced by H2O2 treatment following the reviewer’s suggestions. We have added the description of (i) “endpoints”, which represent the terminal part of cellular ramification and of (ii) “process length”, that indicate the measure of processes elongation (lines 126-133). Moreover, in lines 157-159, 165-167 and 171-175 we have explained that (iii) fractal dimension is an index of cell complexity, (iv) lacunarity is a measure of changes in soma, (v) maximum span across the hull is the maximum distance between two points across the convex hull, (vi) perimeter and (vii) area parameters referring also to the picture of a single cell in Figure 13. The figure caption has been adapted to the adjusted image (lines 458-466).
- There is no evidence that hexarelin could exert a pro-oncogenic effect through the reduction of caspase-3. Even though the canonical mechanism of caspases leads one to think that their inhibition could promote cancer proliferation[4]; recently, non-apoptotic roles of caspases have been identified but are not fully understood[5]. In fact, caspase-3 activation could bind DNA and enhance angiogenesis and chemotherapy resistance[6]. In this work we have focused on the capability of hexarelin to counteract the apoptosis pathway consequent to H2O2-induced damage. Moreover, different reports demonstrate that hexarelin could exert beneficial effects against cisplatin-induced cachexia, accelerating body mass recovery[7].
- We apologize for the mistake. Figure 9 has been modified as suggested, using the same order of graphic bars and western blot.
- For each western blot we used β-actin as a control to verify the loading consistency of proteins. As suggested, we have modified Figure 11 adding the corresponding blot of β-actin for each western blot (p-ERK/t-ERK, p-p38/t-p38 and p-Akt/t-Akt) and the caption has been modified accordingly (line 261).
- Various reports have demonstrated that hexarelin could protect cells from mitochondrial damage, both in vitro and in vivo[8]. In this study, we demonstrated that H2O2 treatment can lead to cell death via activation of the BCL-2 family and through the increase activities of caspases[9]. The activation of the stress-sensitive MAPK pathway shows the direct involvement implication of mitochondrial signalling systems for induction of apoptosis. The ability of hexarelin to modulate mRNA levels of Bax and Bcl-2 and to reduce caspase-3 activation, as shown in this manuscript, suggest that hexarelin protects Neuro-2A cells from H2O2-induced apoptosis by modulating mitochondrial functions and the MAPK pathway.
- We have modified all figure legends that now show “H2O2” as suggested.
[1] Mira et al., «Comparative Biological Study of Roots, Stems, Leaves, and Seeds of Angelica Shikokiana Makino»; Zhao et al., «Neuro-Protective Effects of Aloperine in an Alzheimer’s Disease Cellular Model»; Dong et al., «A Potent Inhibition of Oxidative Stress Induced Gene Expression in Neural Cells by Sustained Ferulic Acid Release from Chitosan Based Hydrogel»; Wu et al., «Role of ataxia-telangiectasia mutated in hydrogen peroxide preconditioning against oxidative stress in Neuro-2a cells»; Xu et al., «Spinosin protects N2a cells from H2O2-induced neurotoxicity through inactivation of p38MAPK».
[2] Sirago et al., «Growth Hormone Secretagogues Hexarelin and JMV2894 Protect Skeletal Muscle from Mitochondrial Damages in a Rat Model of Cisplatin-Induced Cachexia»; Biagini et al., «Beneficial Effects of Desacyl-Ghrelin, Hexarelin and EP-80317 in Models of Status Epilepticus»; Bulgarelli et al., «Desacyl-Ghrelin and Synthetic GH-Secretagogues Modulate the Production of Inflammatory Cytokines in Mouse Microglia Cells Stimulated by Beta-Amyloid Fibrils».
[3] Zhao et al., «Hexarelin Protects Rodent Pancreatic Β-Cells Function from Cytotoxic Effects of Streptozotocin Involving Mitochondrial Signalling Pathways In Vivo and In Vitro»; Avallone et al., «A Growth Hormone-Releasing Peptide That Binds Scavenger Receptor CD36 and Ghrelin Receptor Up-Regulates Sterol Transporters and Cholesterol Efflux in Macrophages through a Peroxisome Proliferator-Activated Receptor γ-Dependent Pathway».
[4] Wang et al., «Ropivacaine Promotes Apoptosis of Hepatocellular Carcinoma Cells through Damaging Mitochondria and Activating Caspase-3 Activity»; Checker et al., «Modulation of Caspase-3 Activity Using a Redox Active Vitamin K3 Analogue, Plumbagin, as a Novel Strategy for Radioprotection»; Iqbal et al., «Combinatorial Effect of Curcumin and Tumor Necrosis Factor-α-Related Apoptosis-Inducing Ligand (TRAIL) in Induction of Apoptosis via Inhibition of Nuclear Factor Kappa Activity and Enhancement of Caspase-3 Activity in Chronic Myeloid Cells».
[5] Zhou et al., «Caspase-3 Regulates the Migration, Invasion and Metastasis of Colon Cancer Cells».
[6] Bernard et al., «Cleaved Caspase-3 Transcriptionally Regulates Angiogenesis-Promoting Chemotherapy Resistance».
[7] Sirago et al., «Growth Hormone Secretagogues Hexarelin and JMV2894 Protect Skeletal Muscle from Mitochondrial Damages in a Rat Model of Cisplatin-Induced Cachexia»; Conte et al., «Cisplatin-Induced Skeletal Muscle Dysfunction»; Bresciani et al., «JMV2894, a Novel Growth Hormone Secretagogue, Accelerates Body Mass Recovery in an Experimental Model of Cachexia».
[8] Sirago et al., «Growth Hormone Secretagogues Hexarelin and JMV2894 Protect Skeletal Muscle from Mitochondrial Damages in a Rat Model of Cisplatin-Induced Cachexia»; Zhao et al., «Hexarelin Protects Rodent Pancreatic Β-Cells Function from Cytotoxic Effects of Streptozotocin Involving Mitochondrial Signalling Pathways In Vivo and In Vitro»; «Growth Hormone Secretagogues Preserve the Electrophysiological Properties of Mouse Cardiomyocytes Isolated from in Vitro Ischemia/Reperfusion Heart | Endocrinology | Oxford Academic»; Baldanzi et al., «Ghrelin and Des-Acyl Ghrelin Inhibit Cell Death in Cardiomyocytes and Endothelial Cells through ERK1/2 and PI 3-Kinase/AKT».
[9] Giménez-Cassina e Danial, «Regulation of mitochondrial nutrient and energy metabolism by BCL-2 family proteins»; Zhao et al., «Hexarelin Protects Rodent Pancreatic Β-Cells Function from Cytotoxic Effects of Streptozotocin Involving Mitochondrial Signalling Pathways In Vivo and In Vitro».
Reviewer 2 Report
Neurodegenerative diseases belong to major healthcare problems nowadays. Despite numerous studies concerning etiology and treatment, no agents efficient in the prevention and therapy are avalaible. Thus, finding new therapeutics in the protection of neural cells is very important. The presented research focuses on the neuroprotective role of hexarelin, one of GHS. The study was correctly planned and performed. The introduction, results and material and methods are interestingly described. However, in my opinion, the discussion should be improved. Now, it describes the results, but the results of other studies concerning oxidative stress and hexarelin or neuroprotection and hexarelin should be discussed. English is good, but some small mistakes should be corrected.
Author Response
Reply to Reviewer 2:
- Following the reviewer's suggestion, we have improved the Discussion section from line 307 to 326.
We have cited published works concerning neuroprotective roles of hexarelin, both in vitro and in vivo[1], and its anti-oxidant effects[2]. These studies, using different pathological models, all support our results concerning anti-apoptotic and neuroprotective roles of hexarelin.
[1] Barlind et al., «The Growth Hormone Secretagogue Hexarelin Increases Cell Proliferation in Neurogenic Regions of the Mouse Hippocampus»; Johansson et al., «Proliferative and Protective Effects of Growth Hormone Secretagogues on Adult Rat Hippocampal Progenitor Cells»; Brywe et al., «Growth Hormone-Releasing Peptide Hexarelin Reduces Neonatal Brain Injury and Alters Akt/Glycogen Synthase Kinase-3β Phosphorylation»; Fetler e Amigorena, «Neuroscience. Brain under Surveillance»; Bulgarelli et al., «Desacyl-Ghrelin and Synthetic GH-Secretagogues Modulate the Production of Inflammatory Cytokines in Mouse Microglia Cells Stimulated by Beta-Amyloid Fibrils»; Griffin et al., «Glial-Neuronal Interactions in Alzheimer’s Disease».
[2] Agbo et al., «Hexarelin Protects Cardiac H9C2 Cells from Angiotensin II-Induced Hypertrophy via the Regulation of Autophagy»; Carson, Hardee, e VanderVeen, «The Emerging Role of Skeletal Muscle Oxidative Metabolism as a Biological Target and Cellular Regulator of Cancer-Induced Muscle Wasting»; Sirago et al., «Growth Hormone Secretagogues Hexarelin and JMV2894 Protect Skeletal Muscle from Mitochondrial Damages in a Rat Model of Cisplatin-Induced Cachexia».
Reviewer 3 Report
This work has been devoted to research into the inhibitory effect of hexarelin on apoptosis in Neuro-2A cells induced by hydrogen peroxide. Hexarelin is a synthetic peptide that exhibits a number of bioactivities. It is a hexapeptide (His-D-2-methyl-Trp-Ala-Trp-D-Phe-Lys-NH2) that stimulates growth hormone release by binding to the growth hormone secretagogue receptor (GHSR). Hexarelin may have direct cardiovascular effects in addition to growth hormone release and neuroendocrine effects. Company Mediolanum Farmaceutici was developing examorelin (hexarelin) for the treatment of somatropin deficiency.
Research by the authors of this paper submitted for review has shown that hexarelin is able to protect Neuro-2A cells from the cytotoxic effects caused by H2O2. To demonstrate this, the authors used many evaluation techniques and methods. The way of presenting the results and their discussion, in my opinion, does not raise any objections, therefore I suggest publishing this manuscript in its present form.
Author Response
Dear reviewer, we thank you for your kind comments on our paper.
Round 2
Reviewer 1 Report
The authors have properly addressed all the questions I made. I think that the paper can be accepted in the present form.